# Brief communication: Seasonal prediction of salinity intrusion in the Mekong Delta

Heiko Apel[1], Mai Khiem[2], Nguyen Hong Quan[3], To Quang Toan[4]

[1]Section Hydrology, GFZ German Research Centre for Geoscience, Potsdam, Germany
[2]National Centre for Hydro-Meteorological Forecasting (NCHMF), Ha Noi, Vietnam
[3]Center of Water Management and Climate Change (WACC), Vietnam National University - Ho Chi Minh city (VNU - HCM), Ho Chi Minh city, Vietnam
[4] Southern Institute of Water Resources Research (SIWRR), Ho Chi Minh city, Vietnam

*Correspondence to*: Heiko Apel (heiko.apel@gfz-potsdam.de)

**Abstract.** The Mekong Delta is the most important food production area in Vietnam, but salinity intrusion during the dry season poses a serious threat to agricultural production and livelihoods. A seasonal forecast of salinity intrusion is required in order to mitigate the negative effects. This communication presents a statistical seasonal forecast model based on Logistic Regression using either the ENSO34 index or Streamflow as predictor. The model is able to predict the salinity intrusion up to 9 months ahead with high skill (ROC scores > 0.8). The model can thus be used operationally as a basis for timely adaptation
and mitigation planning.

## 1 Problem setting

The Mekong Delta (MKD) is the most important food production area in Vietnam, responsible for about 56% of the rice production of Vietnam and 20% of the agricultural exports of Vietnam as a whole (source: General Statistics Office of Vietnam (GSO), http://www.gso.gov.vn). As a low-lying coastal area in a monsoonal climate with distinct wet and dry season it is
naturally prone to salt water intrusion into the river and channel network during the dry low flow season. Sea level rise and climate change aggravate this problem causing more severe, longer lasting, and more frequent droughts, with the consequence of more severe (longer lasting and higher salinity levels) and more frequent salinity intrusions during the dry season (Smajgl et al., 2015). The current agricultural production systems and livelihoods over time adapted to a given intensity of salinity intrusion, but these changes pose a serious threat to the agricultural production and livelihood of the population. A drastic
example of the immense negative effects of a strong salinity intrusion is the dry season of 2015/2016. This unprecedented high salinity intrusion, which manifested by the earliest onset of high salinity levels, the highest observed salinity measurements in many places, the longest duration, and the deepest penetration of saline water in the river system ever observed, caused widespread crop losses throughout the delta. Of the 13 provinces in the Mekong Delta, 9 were affected by severe salinity intrusion, and all provinces were affected by water shortage (Nguyen, 2017). Approximately 400,000 ha of cropland were
subject to saline irrigation water, of which 238,276 ha were paddy rice fields. The salinity intrusion also affected 6,575 ha of

vegetables, 29,277 ha of fruit trees and 79,000 ha of aquaculture, mainly of brackish water shrimp. Figure 1 provides an overview of the land use in the Mekong Delta, as observed and classified in 2010 by satellite remote sensing Leinenkugel et al. (2013). It illustrates roughly the main affected cropping types in the coastal areas of the delta. The overall economic damage amounted to 5,500 billion VND, equivalent to about 236 million US$ and 0.74% of the National Gross Domestic Product of the agriculture, fishery and forestry sectors in 2016 (source: GSO). Furthermore the Vietnamese National Steering Center for Natural Disaster Prevention and Control reported that 194,000 households lacked freshwater for domestic use in the Mekong Delta (VDMA, 2016).

The main damaging effect is the lack of fresh water required for irrigation of rice paddies, but also of fruit orchards and vegetable farming. The crops are damaged or even die either by lack of water, or by the adverse effects of high salt concentration in the irrigation water. Because of these effects, salinity intrusion in the Mekong Delta can also be termed as agricultural drought, following the general definition of agricultural drought as a situation, in which plant water demands cannot be satisfied by water availability (Mannocchi et al., 2004;Mishra and Singh, 2010). By terming salinity intrusion an agricultural drought we hereby extend the definition of water availability from a pure physical, quantitative view to a water quality perspective. This agricultural drought is a serious hazard for large parts of the population, for which irrigation-based agriculture is still the basis of its livelihood. An important factor for the large damages were the lack of appropriate mitigation plans, and a timely early warning of the serious salinity intrusion in order to prepare and adapt the agricultural practices for damage reduction.

However, studies and publications dealing with direct forecasts or early warning of salinity intrusion in the Mekong Delta do not exist. This holds true for both short-term and seasonal forecasts. The presented study is thus a novel work in this regional context, but also beyond. Publications about forecast models of salinity intrusion are rather scarce in general. There are just a few papers dealing with this issue. All of them apply different methods to the method presented in this study. The approaches of other studies are a) hydrodynamic modelling of salinity intrusion and the coupling of these models with meteorological and tidal forecast models (Risley et al., 1993), b) the use of Artificial Neural Networks (Lu and Chen, 2010;Roehl Jr. et al., 2013), c) kernel-based support vector machine (Rohmer and Brisset, 2017), and d) power law models derived from hydrodynamic models (Etemad-Shahidi et al., 2008). The approaches applied in these studies are quite complex and require an extensive set of data and models.

Although there is this clear gap in literature, operational forecasts of salinity intrusion in the Mekong Delta are made by the National Centre for Hydro-Meteorological Forecasting (NCHMF), and the Southern Institute of Water Resources Research (SIWRR) under the authority of the Ministry of Agriculture and Rural Development (MARD). The forecasts are provided to MARD and distributed among governmental agencies and provincial governments in the MKD. Both forecasts are based on a complex chain of hydrological and hydraulic models, which are fed by precipitation and tidal monitoring data and forecasts. The forecasts of NCHMF are short termed, i.e. with a lead time of 10 days and are issued on a regular basis every 5-10 days. The forecasts of SIWRR cover also longer lead times up to a maximum of two months. The core of the forecast model of SIWRR is described in Toan (2014). This lead time is, however, too short to plan and adapt the cropping system well ahead

of the drought event. Forecasts with lead times of several months are required to change the cropping system and to prepare for the crop planting during the dry (December to April) season.

A drought as occurred in 2016 is expected to occur more often in the future, as negative rainfall anomalies occurring with El Niño events are expected to occur more frequently (Azad and Rajeevan, 2016). Moreover, sea levels around the Mekong Delta are rising and are expected to rise further in future (Smajgl et al., 2015). Rising sea levels cause increasing backwater effects

and reverse flow in the river channels, and thus promote salinity intrusion during the dry season. The 2016 event was a wake-up call for the society and officials in Vietnam, as it proved that large structural problems in drought management and mitigation exist. In order to support disaster management this study aims at the development of a reliable and simple salinity intrusion forecast system enabling lead times of several months, and thus a better early warning, and an adaptation to and mitigation of the adverse impacts of salinity intrusion and agricultural droughts in the Mekong Delta.

**2 Hydrology, data and method**

The hydrology of the Mekong Delta and Mekong basin is dominated by a monsoonal climate, separating the hydrological year into distinct rainy/high flow and dry/low flow seasons. The hydrological regime lags the climate regime depending on the location in the basin. In the Mekong Delta this lag is most noticeable due to the time required for transforming rainfall in the about 800,000 km$^2$ large basin to river discharge and routing the discharge to the delta. The lag time in the delta between onset

and end of the rainy and flood season can be up to 2 months. Moreover, the dry season discharge in the Mekong Delta crucially depends on the discharge generated in the Mekong basin, which is in turn depending on the amount of rainfall in the basin during the monsoon period, i.e. discharge in the Mekong basin and delta is highly correlated to the monsoon intensity (Delgado et al., 2012;Räsänen and Kummu, 2013). The SE-Asian monsoon intensity is itself determined by the periodically changing sea surface temperatures in the western central Pacific Ocean (West Pacific Warm Pool), associated with the El Niño Southern

Oscillation (ENSO). Of particular importance for the monsoon strength is the situation of ENSO in winter and spring prior to the monsoon season, when the general circulation and moisture fluxes for the monsoon season are initiated (Ju and Slingo, 1995). Strong events of the South-East Asia monsoon are associated with La Niña events, while weak monsoons and thus higher chances of salinity intrusion in the dry season are associated with El Niño events (Ju and Slingo, 1995). Therefore, a general causal chain for dry season discharge and thus salinity intrusion in the Mekong Delta can be formulated as follows:

ENSO determines the intensity of the SE-Asian monsoon, the monsoon intensity determines the rainfall amount over the Mekong basin, the rainfall amount determines the flood season discharge, the flood season discharge is itself indicative for the following dry season discharge, and the dry season discharge controls the salinity intrusion in the Mekong delta.

This general causal chain forms the basis for the simple salinity intrusion forecast model presented in this study. Firstly, an early forecast will be attempted utilizing an ENSO index as predictor. Secondly, an additional forecast will be tested using

flood season and early dry season discharge as predictor. Monthly ENSO indexes were collected from the Physical Sciences Division (PSD) of the Earth System Research Laboratory (ESRL) of the National Oceanic and Atmospheric Administration

(NOAA) (https://www.esrl.noaa.gov/psd/data/climateindices/list/). Furthermore, monthly discharge data were collected from the Southern Regional Hydrometeorological Center (SRHMC) for the gauging station Tan Chau, located at the Mekong (Tien in Vietnam) branch of the Mekong in the Delta (Figure 1). The salinity of surface water in the MKD is also measured by SRHMC, but during the dry season only. Salinity is monitored at 39 locations in the MKD, and is determined by collecting water samples in mid-river at 0.2, 0.5 and 0.8 of the water depth. The samples are analysed in the laboratory, and the reported salinity is the mean of these three measurements. If the water depth is below 3 m, only one sample is taken at 0.5 of the depth. Due to constraints in personal and monetary resources, the monitoring is, however, not time continuous. The general scheme is to measure 2-3 days in a row at 2 hours intervals (i.e. 12 measurements per day). This measurement period is followed by a 2-4 days without measurements, after which the monitoring resumes. The salinity time series used in the model development was recorded at gauge Son Doc in Ben Tre province in the estuary of the Mekong (Tien) river branch. Son Doc is located about 175 km downstream of the gauge Tan Chau, and about 24 km upstream of the river mouth (Figure 1). The salinity measurements covered all dry seasons in the time span 1996 – 2016. The temporal coverage of the salinity measurements at Son Doc is shown in Supplement S1. In order to derive a meaningful predictand of salinity intrusion, the mean salinity of February and March (FebMar) was calculated from the available salinity measurements. This aggregation time period was chosen, because it coincides with the vegetative stage of irrigated paddy grown during the dry season, which is the most sensitive phase of paddy rice to high salinity levels (Zeng et al., 2001).

The envisaged seasonal forecast of salinity intrusion does not aim at forecasting the exact mean salinity intrusion of this period, but rather at forecasting the probability of exceedance of critical levels of salinity intrusion. For paddy rice a salinity of the irrigation water exceeding 4 g/l is seen by the authorities in Vietnam as too high for the plants to survive during the vegetative stage (compare Zeng and Shannon, 2000). Therefore a mean salinity of 4 g/l during February and March is adopted as critical salinity level for the forecast model. The mean salinity threshold during this period means that this threshold will be exceeded at 43% of the time, considering the negative exponential distribution of the data (cf. Supplement S2), and assuming that the discontinuous measurements are representative for the whole time period. In addition to the critical salinity level a threshold of 3 g/l mean FebMar salinity is used as predictand. This threshold is exceeded at 53% of the time in February and March (cf. Supplement S2) and serves as a warning threshold, indicating a strong salinity intrusion with chances of salinity also exceeding 4 g/l at times. Rice irrigated with this salinity threshold might survive depending on the duration of the irrigation with saline water, but losses in crop yield have to be expected (Grattan et al., 2002;Zeng and Shannon, 2000).

The forecast model is based on a Logistic Regression (LR), i.e. a linear statistical model that relates categorized values to continuous, real-number type predictors (Menard, 2009). LR has to our knowledge never been used in forecasting salinity intrusion. Moreover, a seasonal forecast of salinity intrusion, i.e. a forecast with several months lead time, has not been published before. The presented work is thus also novel in this aspect. Compared to the published approaches of salinity intrusion listed in the introduction and the model chains used for the operational forecasts, forecasting with LR is a very simple and data-sparse approach.

The categories to be predicted by the LR are the mean FebMar salinity values categorized in bins above or below the defined salinity thresholds (4 g/l and 3 g/l). The continuous predictors are monthly ENSO indexes and Standardized Streamflow Indices (SSI) values. For this kind of regression LR is the appropriate tool. LR is very flexible in its application, because it is not limited to normally distributed predictors, as e.g. the possible alternative method, the Linear Discriminant Analysis (Pohar et al., 2004). Regression models were developed with a single predictor using either ENSO indexes or SSI, following the causal

chain leading to salinity intrusion in the MKD described above. The ENSO indexes tested were monthly ENSO1, ENSO3, ENSO4, and ENSO34 indexes. All of these indexes aim at representing the state of the El Niño Southern Oscillation by considering sea surface temperatures at different regions of the Pacific Ocean. Among these indices ENSO34 is regarded as the most appropriate sea surface temperature index representing the general state of the ENSO (Bamston et al., 1997). The testing of different indexes aims at the identification of the most robust predictor for salinity intrusion in the MKD. All of the

ENSO indexes used in the forecast models start in April of the year before the dry season, i.e. with a lead time of up to 9 months before the start of the forecasted FebMar time period. For the streamflow predictors, the monthly discharges recorded at station Tan Chau were transformed into SSI. This transformation is similar to transforming precipitation records into Standardized Precipitation Indexes (SPI), as typically done in drought studies and drought definitions (Mishra and Singh, 2010). SSI has been applied as predictand in seasonal forecast studies, e.g. for predicting streamflow in Southern Africa

(Seibert et al., 2017). SSI normalizes the monthly discharges by fitting a Gamma distribution to the observed long-term record of discharges (here from 1980 – 2016) and transferring it in a normal distribution with a mean of 0. An SSI value of 0 indicates thus the normal hydrological state, while negative values indicate a water deficiency. SSI has the advantage that a drought condition can be directly recognized by the SSI value, and that the prediction models can be easier transferred and compared to other gauging station with different streamflow magnitudes. Another strength of SSI is that it can be calculated for a variety

of time scales. This is important, because droughts usually manifest over extended time periods. SSI was thus derived for different time scales ranging from 1 month (SSI1) to 6 months (SSI6), each starting in June prior to the dry season, i.e. with a maximum of 7 months lead time. The calculation of SSI was performed with the R-package SPEI (Vicente-Serrano et al., 2012). Supplement S3 provides a list of all predictors used in the detection of the best performing forecast models.

The Logistic Regression models were fitted by iteratively reweighted least squares using the R-function *glm* for a binomial

model type, which represents the Logistic Regression. A fitted LR model provides estimates of the probability of exceedance of the defined salinity thresholds depending on the predictor value. One model was fitted for all predictors and lead times, and the best performing ENSO and SSI predictors for the different lead times were manually selected according to the following criteria:

- The Receiver Operator Characteristic (ROC) score (Mason, 2008).

- The Akaike Information Criteria (AIC) (Burnham and Anderson, 2004).
- The Cragg and Uhlers (also known as Nagelkerke) Pseudo-$R^2$ (Nagelkerke, 1991), which is defined for categorical variables analogously to the normal $R^2$ for continuous variables.
- The accuracy (rate of correct forecasts).

In order to test the robustness of the linear models a Leave-One-Out Cross validation (LOOCV) was also performed, as applied
in Apel et al. (2018) for forecasting seasonal streamflow in Central Asia. A robust model is characterized by a model, for
which the performance values of the LOOCV do not deteriorate compared to the performance of the model using the full data
set. Therefore, the ROC scores and accuracies of the LOOCV were used in addition to the performance criteria listed above
for the selection of the best performing forecast models.

## 3 Results and discussion

The performance testing of the ENSO predictors identified the ENSO34 index as best performing ENSO index. The best
forecast could be obtained with the April index, i.e. with a lead time of 9 months. Figure 2 ((a) and (b)) illustrates the
performance of the forecast model. Using only the ENSO34 index of April a ROC score of 0.98 and 0.8 and an accuracy of
95% and 71% could be achieved for the 3 g/l and 4 g/l thresholds, respectively. The pseudo-$R^2$'s of 0.89 (3 g/l threshold) and
0.4 (4 g/l threshold) confirm this excellent performance, considering the typically lower values of pseudo-$R^2$ compared to
normal $R^2$ values. This performance is exceptionally high for a seasonal forecast with such a long lead time. The LOOCV
resulted in similar high performances, thus indicating the robustness of the statistical model. The logistic regression curves in
the top-right insets show that the 3 g/l threshold can be very well discriminated by the ENSO34 index. The steep slopes of the
probabilities changing from drought classification to non-drought classification illustrate this. For the 4 g/l threshold the slopes
are more gentle, indicating a less pronounced discrimination, which is expressed by the lower performance values. The bottom
insets in the figure panels show the observed drought events with reported high salinity intrusion and the forecasts of the
model, including the LOOCV forecasts. It can be seen that only 1 out of 21 dry seasons would have been misclassified as
drought for the 3 g/l threshold. For the 4 g/l threshold 6 out of 21 events would have been wrongly predicted. In this context it
has to be noted that the probability threshold for classifying a forecast as droughts was set to 0.5. Using different probability
thresholds for drought classification has been tested and resulted in different classification errors, but the number of wrong
classifications and thus the performance values remained the same.
The forecast performance with ENSO is decreasing with decreasing lead times, as shown in Figure 3(a) by decreasing ROC
scores and increasing AIC values. This finding is in line with the causal chain explained above, where ENSO is preceding the
monsoon development. During the monsoon period ENSO changes already to a different state, but having less impact on the
monsoon intensity (Ju and Slingo, 1995), thus the value of ENSO as predictor for the salinity intrusion is decreasing.
An opposite behaviour is observed for the SSI predictors (Figure 3(b)). These show in general an increasing performance with
decreasing lead time. This behaviour also reflects the hydrological system of the Mekong, where the streamflow at the late
flood (October-November) and early dry season (December) are an aggregated measure of the total monsoonal rainfall amount
over the Mekong basin, which is more reliable compared to streamflow during the early and high flood season. The SSI3
predictor, i.e. an aggregated index of three months of discharge, performs best, but only slightly better than SSI4 and SSI6 (not
shown). The best forecasts were obtained for November and December, i.e. with 1 – 2 month(s) lead time. Interestingly the

SSI forecasts were in general better for the 4 g/l threshold than for the 3 g/l threshold (Figure 3), which is opposite to the forecasts with ENSO. The 4 g/l threshold exceedance can be forecasted with a ROC score of 0.85 and an accuracy of 85% with SSI3 in December and November (Figure 2 (c) and (d)). Only 3 out of 21 events were wrongly classified for this salinity threshold. This means that the early salinity intrusion forecast by ENSO for the critical salinity threshold of 4 g/l can be further improved with SSI forecasts a few months prior to the dry season, which is important for the actual implementation of disaster mitigation plans.

However, in order to obtain a continuous, well performing forecast model, the forecasts during the flood season (July-October) need to be improved. Suitable predictors during this period would be rainfall estimates over the Mekong basin, following the causal chain of salinity intrusion in the MKD outlined in the method section. These rainfall predictors could be derived from the real-time monitoring rainfall network in the Mekong basin, or near-real time satellite-based rainfall products, as e.g. the TRMM-based Multi-satellite Precipitation Analysis (TMPA/3B4x) with its latency of 1-2 month. This needs to be tested during further developments of the model.

## 4 Conclusions and outlook

The proposed simple linear seasonal forecasting model of salinity intrusion in the Mekong Delta based on ENSO and SSI predictors proved to be a useful tool for an early warning of salinity intrusion during the dry low flow season in the Mekong delta. The exceedance and non-exceedance of critical and high levels of salinity at the Son Doc gauging station could be forecasted with high probabilities. Combining the ENSO and SSI forecast models results in a forecasting system that could deliver an early warning as early as 9 months prior to the period of the dry season most critical for paddy rice, but also for other crops and fruit orchards. The early forecasts with ENSO in April before the actual flood and the following dry season could serve as an early warning of a likely high salinity intrusion. The later forecasts based on SSI would then provide more reliable forecasts of a severe salinity intrusion that would cause high damages and negative impacts of the livelihood of the population in the coastal provinces of the Mekong Delta, if no mitigation actions are initiated in time. Based on the long lead times of the forecasts appropriate mitigation measures can be planned already during the flood season, i.e. well ahead of the dry season, and then activated if the early forecasts are confirmed by the late forecasts based on SSI, i.e. actual observed discharges. The proposed forecasting model could thus be used as a data-based support for disaster mitigation planning. However, it could be further improved to obtain robust forecasts during the flood season, i.e. lead times of 5-7 months, by testing rainfall estimates over the Mekong basin as predictor.

Due to its simplicity the model can easily be transferred to other gauging stations in the Mekong Delta. This requires mainly sufficiently long time series of salinity measurements, because long term discharge records for the calculation of SSI are readily available for the main gauges in the VMD, and the time series of ENSO indexes are available from public sources. As a rule of thumb, salinity time series of about 15 years and longer should be sufficient for fitting the model (cf. Apel et al., 2018). While the studied gauge Son Doc can be regarded as representative for the general salinity intrusion in the Mekong

Delta, forecasts of a larger number of stations would increase the data-based evidence of an imminent severe salinity intrusion affecting the whole delta. Moreover, due to the simple model structure and data requirements, the model could be applied beyond the Mekong delta in other coastal areas draining larger basins in a monsoonal climate, where similar hazards of dry season salinity intrusion exist.

After the model development, the model was validated by forecasting the salinity intrusion in the dry season 2020. It predicted the observed high salinity intrusion with high confidence using both the ENSO34 predictor of April 2019 and the SSI3 predictors of November and December 2019. This means, that the observed severe salinity intrusion in the dry season of 2020 could have been predicted with a lead time of 9 months. This lead time is certainly sufficient for timely mitigation and adaptation planning. The proposed model can therefore provide a data-based support for decisions of the Vietnamese government for the future salinity intrusions during the dry seasons. This could avoid severe damages and negative impacts as occurred in 2015/2016, when no preventive and mitigating actions were taken. Mitigation actions should include not only local measures in the VMD, but also negotiations with the Mekong riparian countries with the aim of adapting the operation schedule of reservoirs in the Mekong basin in order to maintain sufficient flow during the dry season. Particularly this aspect needs long term preparation, both in terms of reservoir operation planning and for the required political discussions among the riparian countries.

**Data availability.** Data are available upon request from the corresponding author (heiko.apel@gfz-potsdam.de).

**Author contributions.** HA developed the method and wrote the original draft of the paper. All authors contributed to the preparation of this paper.

**Competing interests.** The authors declare that they have no conflict of interest.

**Acknowledgements.** This study was performed as part of the research project Catch-Mekong (https://catchmekong.eoc.dlr.de/) and the project "Transforming agricultural livelihoods for climate change adaptation in the Vietnamese Mekong Delta: A case study in Ben Tre province". Funding was provided by the German Ministry of Education and Research (BMBF, FKZ: 02WM1338C) and by the Vietnamese Ministry of Science and Technology (MOST, grant number: KHCN-TNB-DT/14-19/C20).

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

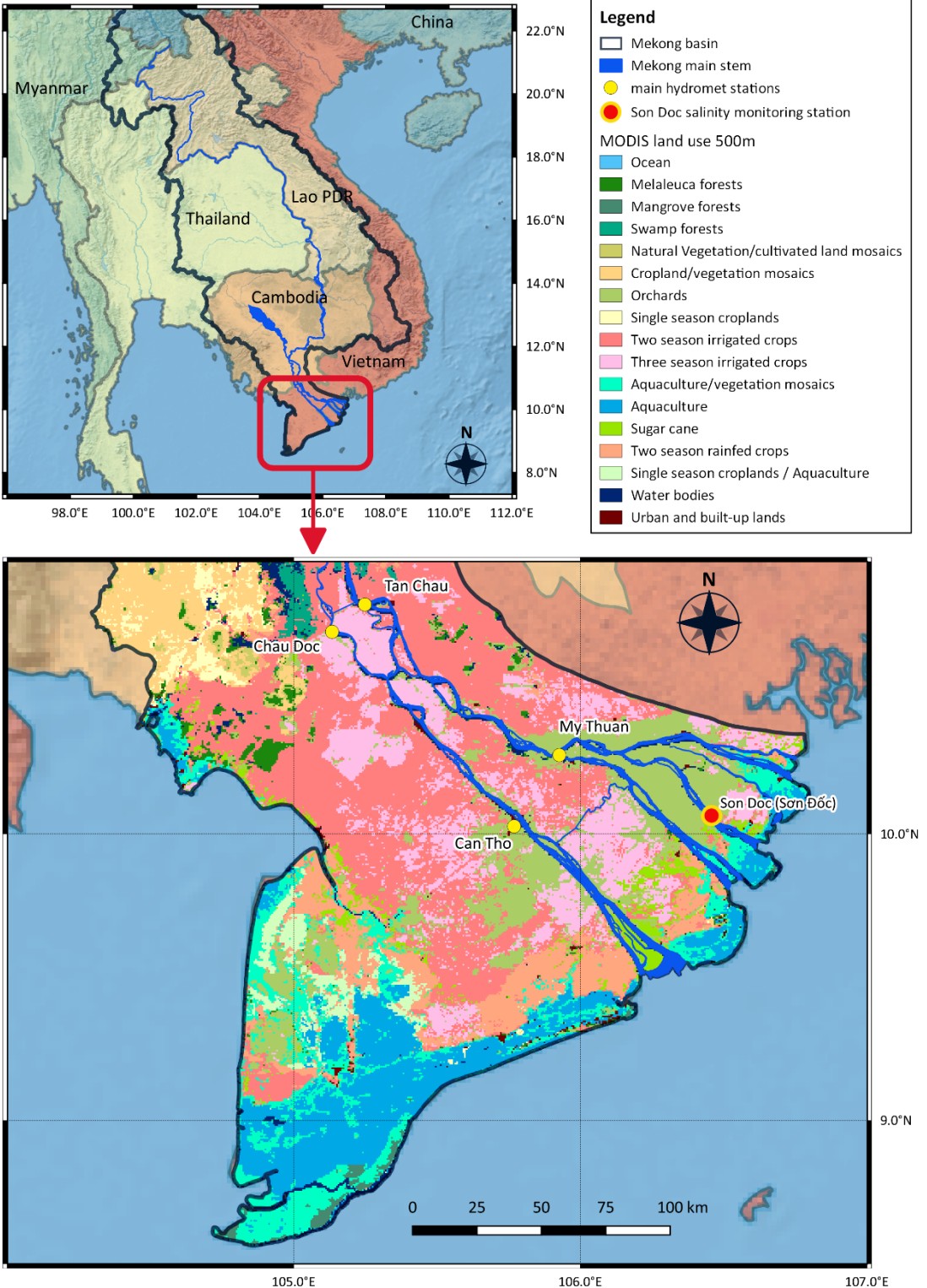

 **Figure 1: Overview map of the study area Mekong Delta. Top left: Regional overview showing South-East Asia with the Mekong basin and delta, and the neighbouring countries. The background country, ocean and topography maps are made with Natural Earth (Free vector and raster map data @ naturalearthdata.com). Bottom: The Vietnamese part of the Mekong Delta with the location of the main official and permanent hydro-meteorological monitoring stations and the salinity monitoring station Son Doc. The land use map of the Mekong Delta as in 2010 is shown as reference illustrating the different land use types in the different**
 **regions of the delta. The map was derived at 500m resolution from Moderate Resolution Imaging Spectrometer satellite (MODIS) images (Source: Catch-Mekong Knowledge Hub, https://catchmekong.eoc.dlr.de/Elvis/, provided by the German Aerospace Center DLR. The method of land use classification is described in Leinenkugel et al. (2013)).**

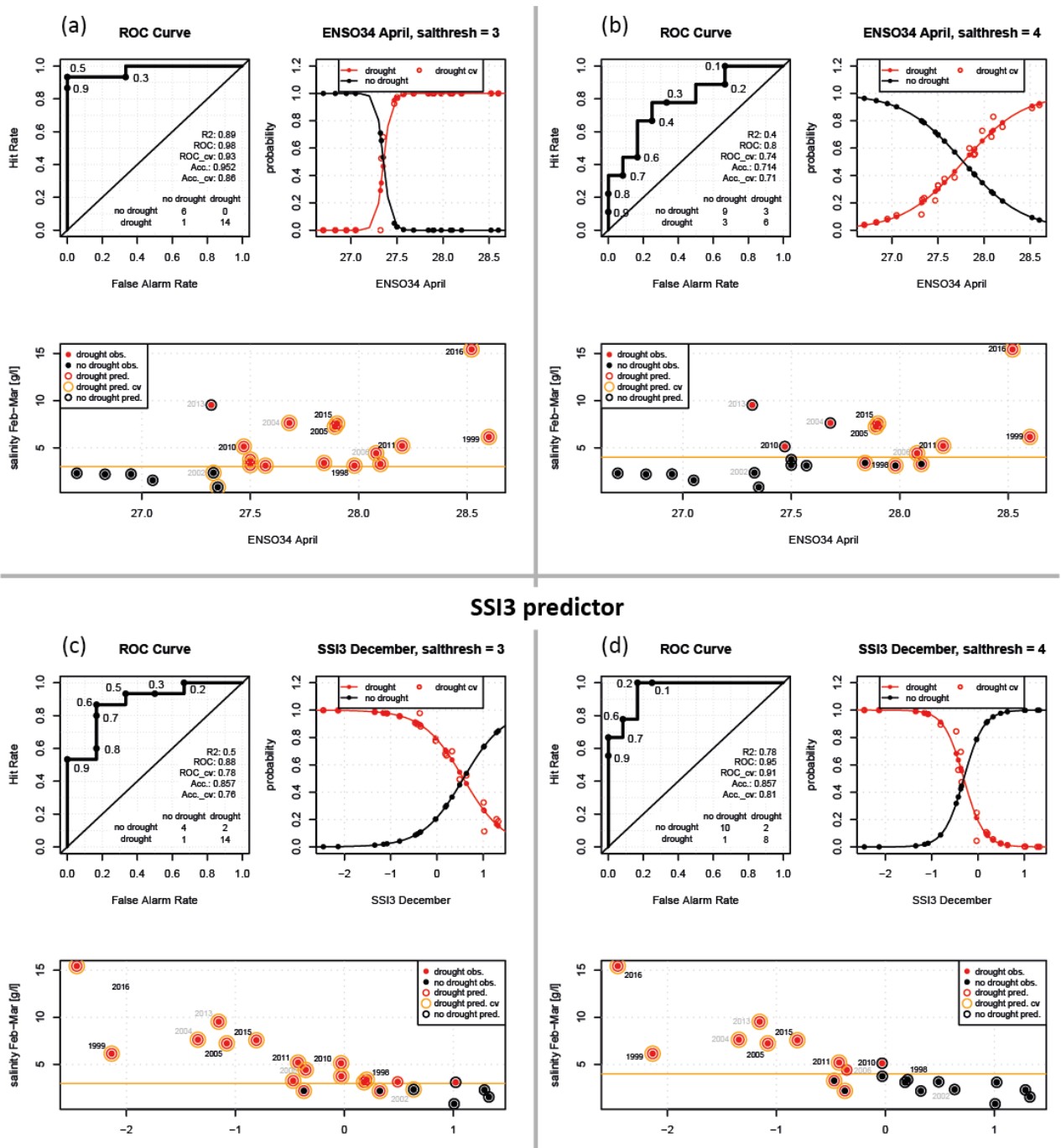

**Figure 2: Best prediction models using ENSO34 index (top row: (a) and (b)) and SSI3 (bottom row, (c) and (d)) as predictors for forecasting exceedance of the 3 g/l (left, (a) and (c)) and 4 g/l (right, (b) and (d)) salinity threshold ("salthresh"). The top-left insets of (a) to (d) show the ROC curves with the following performance criteria: R2 = Cragg & Uhlers pseudo-R², ROC = ROC score, ROC_cv = ROC score of the LOOCV, Acc = accuracy (fraction correct predictions), Acc_cv = accuracy of the LOOCV. Top-right**

 insets of (a) to (d) show the logistic regression results with the probabilities of exceedance and non-exceedance of the salinity thresholds in dependence of the predictor. The bottom insets of (a) to (d) show the observed mean February-March salinity levels at Son Doc and the predictions in terms of exceedance of the salinity thresholds.

335

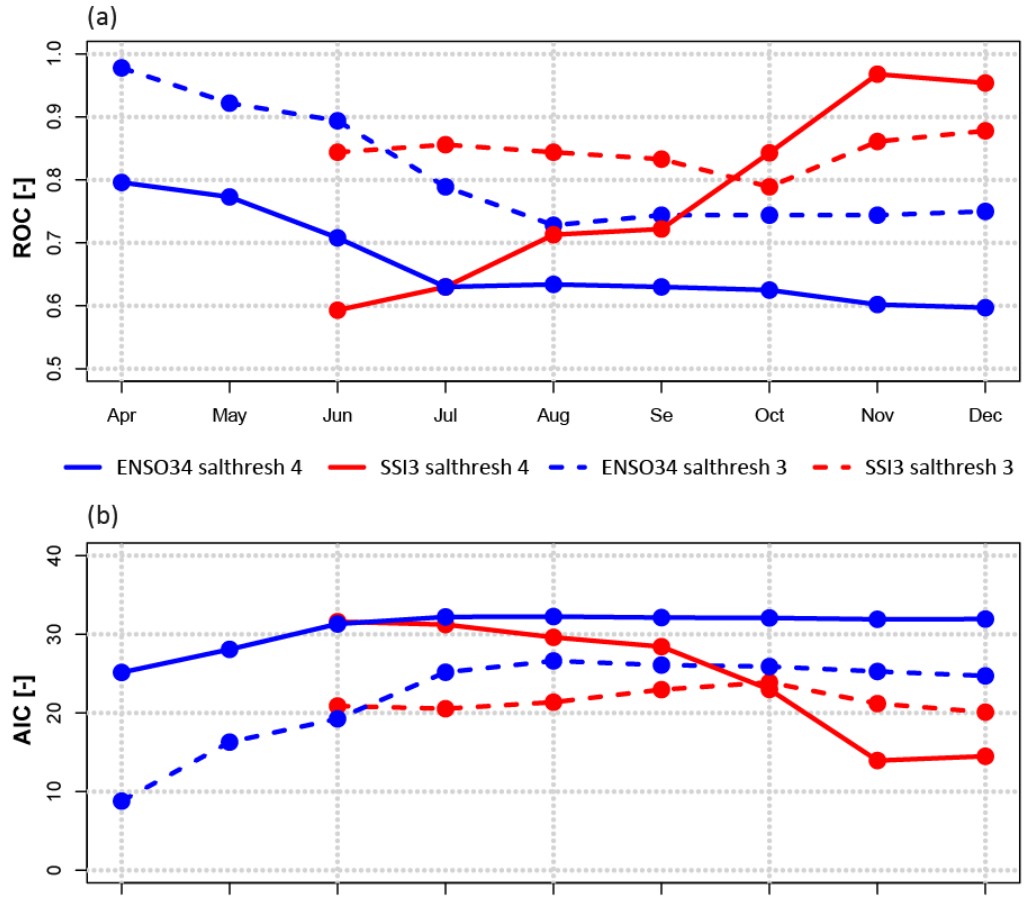

Figure 3: Performance of logistic model with ENSO34 and SSI3 predictors at different lead time in terms of (a) ROC score, and (b) AIC. The months of the x-axis denote a forecast at the end of the indicated month prior to the dry season. For the mean February-March predictand a forecast in December means 1 month lead time for the predictand season, in April a lead time of 9 months.

340