# Peer review of "Brief communication: Seasonal prediction of salinity intrusion in the Mekong Delta"

_Natural Hazards and Earth System Sciences, 2019_

## Referee Comment (RC1) · Anonymous Referee #1 · 4 Feb 2020

This communication presents a simple statistical seasonal forecast model to predict the salinity intrusion with a leading time of 9 months. The model is expected to be used as a basis for timely adaptation and mitigation planning, which is urgently needed for the imminent severe salinity intrusion. That is, its outputs serve as a reference for negotiations with the riparian countries to adapt the operation schedule of reservoirs in the Mekong basin to maintain sufficient flow during the dry season for mitigation of the impacts of the expected very low dry season flow in the year to come, as well as sharing the operation information to downstream countries for mitigation planning. The manuscript would become meaningful when the proposed forecast model becomes an operational tool.

1. English language must be significantly improved before the manuscript can be con-

sidered for acceptance. For example, one sentence in the abstract contains multiple grammatical mistakes: "This communication present a simple statistical seasonal forecast model able to predict the salinity intrusion up to 9 months ahead with high skill." The authors are requested to properly take care of the English writing throughout the whole manuscript.

2. Use of in-situ measurements of soil intrusion is considered in the forecast model, while it is rather time-consuming and cost-ineffective. Would the authors comment on the use of remotely-sensed salinity intrusion (e.g. https://doi.org/10.1186/s40645-019-0311-0) in the forecast model?

3. The description about the salinity intrusion is rather comprehensive in the current version of the manuscript. It can be significantly shortened. In contrast, how the proposed forecast model is formulated and works are much less described and thus must be properly enhanced.

4. Figure 1 shows the land use over the Mekong Delta, while it presents the 2010 status. How is the land use over the Mekong Delta changing with time? How does the evolving land use influence on the proposed forecast model?

5. How the human-made disturbance impacts on the water flow from upstream to downstream along the Mekong River should be addressed. The current manuscript only concerns with the impacts of natural disturbance, i.e. climate. Unfortunately, many reservoirs have been constructed over the upstream and have significantly modified the water flow. How such a human-made disturbance factor influences the performance of the proposed forecast model should be clarified before the model can be used for the operational purpose.

---

## Author Comment (AC1) · 19 Feb 2020

Reply to reviewer comment RC1:

We thank the reviewer for his/her efforts to evaluate the quality and potential impact of our manuscript. We have read the comments carefully and reply to each comment individually below. The original reviewer comments are marked in blue.

1. English language must be significantly improved before the manuscript can be considered for acceptance. For example, one sentence in the abstract contains multiple grammatical mistakes: "This communication present a simple statistical seasonal forecast model able to predict the salinity intrusion up to 9 months ahead with high skill." The authors are requested to properly take care of the English writing throughout the whole manuscript.

Thanks for pointing this out. The language will be double-checked in the revised version.

2. Use of in-situ measurements of soil intrusion is considered in the forecast model, while it is rather time-consuming and cost-ineffective. Would the authors comment on the use of remotely-sensed salinity intrusion (e.g. https://doi.org/10.1186/s40645-019-0311-0) in the forecast model?

In-situ measurements were used, because surface water salinity is measured operationally by the hydro-meteorological service at various locations in the Mekong Delta. The hydro-meteorological service is also responsible for the official forecast of the salinity intrusion in the Mekong Delta. The data is thus readily available at the responsible agency, and therefore it is straight-forward to use this data set for our prediction model. However, it would be of course useful to get additional data based on remote sensing platforms, because remote sensing could provide a better overview on the spatial dimension of the salinity intrusion. If remote sensing can deliver reliable measurements of surface water salinity, I would certainly support the use of it. However, we cannot judge whether this is possible or not, as this is not our field of expertise. The cited paper covers the determination of soil salinity, which is not the target of our forecast, but rather the outcome of saline water used for irrigation.

3. The description about the salinity intrusion is rather comprehensive in the current version of the manuscript. It can be significantly shortened. In contrast, how the proposed forecast model is formulated and works are much less described and thus must be properly enhanced.

We think that the comprehensive description of the salinity intrusion in the Delta and their negative consequences is required to illustrate the necessity of a reliable seasonal forecast and to motivate the research presented in the manuscript. However, we agree that the method could be described in more detail, even though all applied methods are rather standard. In order to stay within the limits of a brief communication, we propose to provide more details in a supplement. We would ask the editor to give a statement about this suggestion. If a supplement is not favoured, we would shorten the motivation part and provide an additional paragraph about the methodological details.

4. Figure 1 shows the land use over the Mekong Delta, while it presents the 2010 status. How is the land use over the Mekong Delta changing with time? How does the evolving land use influence on the proposed forecast model?

The land use in the Mekong Delta is changing quite dynamically as a consequence of economic, political and environmental pressures. In the coastal regions the salinity intrusion is one of the main drivers of land use change. If sufficient amounts fresh irrigation water are not available, farmers and the provincial administration adapt to this situation by e.g. growing more saline tolerant crops, or by changing the farming systems. If the latter aspect this is supported by governmental incentives, as

e.g. in Soc Trang province, where the replacement of paddy rice crops by saline or brackish shrimp production was officially supported, significant changes in land use can occur in a short time. The map in figure 1 is thus mainly for illustrative purpose, showing the different land use in the coastal regions compared to more upstream areas, and to highlight which main land use types are mainly affected by salinity intrusion. In this context it is noteworthy that the differences in land use in the Mekong delta are mainly governed by salinity intrusion and different inundation dynamics.

However, changing land use in the coastal region does not influence the forecast model. The salinity intrusion is dominated by the interplay of tidal forces and river discharge during the low flow season (see e.g. Dang et al., 2019). Both factors are not influenced by land use in the coastal region. The dry season discharge can be altered by storage of flood water during the flood season in high-dike compartments in the upper parts of the Vietnamese Mekong Delta and its release during the early stages of the dry season, but only to a very limited extent (Thanh et al., 2020; Triet et al., 2017). Therefor even the hypothetical conversion of all flood compartments into areas protected by high dikes (i.e. a large scale land use change in the upper part of the Vietnamese part of the delta) would hardly impact on the salinity intrusion and thus the applicability of the proposed model.

5. How the human-made disturbance impacts on the water flow from upstream to downstream along the Mekong River should be addressed. The current manuscript only concerns with the impacts of natural disturbance, i.e. climate. Unfortunately, many reservoirs have been constructed over the upstream and have significantly modified the water flow. How such a human-made disturbance factor influences the performance of the proposed forecast model should be clarified before the model can be used for the operational purpose.

We agree to the statement that man-made disturbances, particularly the ongoing and planned development of hydropower dams, have a considerable impact on the hydrological regime of the Mekong, and thus also on the salinity intrusion. The dam development causes numerous problems ranging from shifts of the hydrological regime, reduced sediment delivery and thus problems for the morphological stability of the delta, disruption of the river ecological system and negative consequences for the Mekong fishery and livelihood of many people. With regard to salinity intrusion the dam induced shift of the hydrological regime towards lower flood season discharge and higher dry season discharge, it can be stated that the dams could partly alleviate the foreseen negative impacts of the rising sea levels on salinity intrusion. This would, from our point of view, be the only potential benefit of the dam development in the Mekong basin for the Mekong delta. This benefit is, however, compromised by the reduction of sediment delivery caused by dams. This causes a deepening of the river channels in the delta, which in turn causes a higher salinity intrusion (Hackney et al., 2020; Tu et al., 2019; Jordan et al., 2019). Moreover, as this potential benefit is subject to political and economic decisions taken in the upstream countries, it is not a reliable mitigation measure for salinity intrusion in the Delta.

Due to the complex nature of this subject, we would thus refrain from discussing the particular impacts of the dam development in the manuscript. Including such a discussion would surely go beyond a brief communication, and there are numerous papers out illustrating the negative effects of hydropower development in the Mekong basin. But we would include the dam issue as a general factor in the discussion, because if the hydrological regime will substantially change due to human interference, the presented forecast models needs to be re-fitted to the new regime, and potentially the operation or actual storage volume of the dams need to be considered as co-variate in the prediction models.

**References**

Dang, V. H., Tran, D. D., Pham, T. B. T., Khoi, D. N., Tran, P. H., and Nguyen, N. T.: Exploring Freshwater Regimes and Impact Factors in the Coastal Estuaries of the Vietnamese Mekong Delta, Water, 11, 782, 2019.

Hackney, C. R., Darby, S. E., Parsons, D. R., Leyland, J., Best, J. L., Aalto, R., Nicholas, A. P., and Houseago, R. C.: River bank instability from unsustainable sand mining in the lower Mekong River, Nature Sustainability, 10.1038/s41893-019-0455-3, 2020.

Jordan, C., Tiede, J., Lojek, O., Visscher, J., Apel, H., Nguyen, H. Q., Quang, C. N. X., and Schlurmann, T.: Sand mining in the Mekong Delta revisited - current scales of local sediment deficits, Scientific Reports, 9, ARTN 17823, 10.1038/s41598-019-53804-z, 2019.

Thanh, V. Q., Roelvink, D., van der Wegen, M., Reyns, J., Kernkamp, H., Van Vinh, G., and Linh, V. T. P.: Flooding in the Mekong Delta: the impact of dyke systems on downstream hydrodynamics, Hydrol. Earth Syst. Sci., 24, 189-212, 10.5194/hess-24-189-2020, 2020.

Triet, N. V. K., Dung, N. V., Fujii, H., Kummu, M., Merz, B., and Apel, H.: Has dyke development in the Vietnamese Mekong Delta shifted flood hazard downstream?, Hydrol. Earth Syst. Sci., 21, 3991-4010, 10.5194/hess-21-3991-2017, 2017.

Tu, L. X., Thanh, V. Q., Reyns, J., Van, S. P., Anh, D. T., Dang, T. D., and Roelvink, D.: Sediment transport and morphodynamical modeling on the estuaries and coastal zone of the Vietnamese Mekong Delta, Continental Shelf Research, 186, 64-76, https://doi.org/10.1016/j.csr.2019.07.015, 2019.

---

## Referee Comment (RC2) · Anonymous Referee #2 · 21 Feb 2020

The authors provided an interesting manuscript on a topic that has strong relevance for actual societal problems in Vietnam and likely beyond. A seemingly novel method for long-term forecasting of salt water intrusion in cultivated lowland areas is presented, which could provide useful early warning information for damage control in agricultural production. Statistical tests by the authors result in good confidence of model performance, leading to recommendations for wider application. However, a precise idea of the actual added value of the proposed model is not communicated clearly enough, due to several reasons. These are discussed in detail below, but can be summarised as a lack of description of similar existing models, the description of input data used, and the limited possibility for model adoption due to a limited description of the model itself and data requirements. In addition, certain aspects of style, grammar, accuracy of statements and embedding in literature should all be improved in order to achieve an appropriate quality for scientific publishing with this high-profile journal.

Nonetheless, the reviewer believes there is strong potential in the manuscript (especially due to the apparent societal demand for the model); and as such, a major revision is recommended with strong encouragement for follow-up by the authors. In order to allow improvements on the remarks made by the reviewer, a long but practical list of suggestions are provided in this documents (general and specific comments), as well as in the marked manuscript document (single-word suggestions).

General comments: The abstract is rather concise, and although this can be appropriate regarding the total length of the article, perhaps a few pieces of information could be inserted. For instance, the authors could improve the technical aspect of the abstract by briefly describing the type of data that predictions are nested in (i.e. drought or ENSO indices), or by providing some quantification to support the claim for "high skill".

The introduction section (chapter 1) clearly emphasises the importance of forecast models with an extended lead time (i.e. months rather than weeks). However, it is unclear whether such models already exist, and thus what is the novelty of the existing work. The authors should dedicate a few lines nested in scientific references to clarify this point, and thus to justify the relevance of their own contribution. In general, the use of literature is quite marginal in the manuscript, and embedding the proposed research in the scientific context is an integral part of scientific writing.

The description of methodology (chapter 2) deserves some critical attention to ensure an appropriate description of processes, used data, and analysis methods. The reviewer refers to the marked manuscript as well as the specific comments provided below for all of these points.

With regards to the results (chapter 3), the authors seem to present a robust set of statistical testing for findings optimal predictors. In the final lines of this chapter, an

[Figure]

interesting point is made about the validity of ENSO-based predictions (optimal on long-term) and SSI-based predictions (optimal for short-term). Was any performance testing done where the two indices were combined, as an "optimised predictor"? If not, the authors may discuss the possibilities for this in future explorations. In addition, chapter 3 in its current form does not provide any discussion with regards to previous scientific works (e.g. regarding other long-term forecasting models), but is mostly restricted to "results".

In the conclusions section, potential application of the proposed model is well described and its wider use is encouraged. However, the requirements with regards to data availability are not entire clear. The authors mention that data availability should be "sufficient", but do not specify or quantify what is the required coverage of flow data and the expected impact on prediction accuracy. This actually links back to the methodology section of the manuscript, where a quantification of data coverage in the presented study is missing as well. More clarity is required on this topic, both in described methodology and in recommendations for future applications.

Specific comments: - Page1,Line7: While acknowledging the Mekong Delta as the most important Vietnamese food production area, the value of this zone with regards to agriculture and food security could be more strongly emphasised by adding two pieces of information: (1) the fraction of rice production out of total (staple) food production in Vietnam; (2) the importance of "nationally produced" food vs. imported food with regards to food security (or possibly exported value). A second line including such information would create a more solid argument as to the context of salt water intrusion and its negative impacts.

- P1,L19: Is indeed the frequency of droughts, rather than the likely duration of the most severe drought (period), the major manifestation of climate-induced intrusion?

- P1,L20: "... agricultural production system and peoples livelihood developed over historical periods and thus adapted to normal intensity of salinity intrusion (...)". This sentence reads slightly unclear and could likely be simplified, e.g. as "…agricultural production systems and livelihoods over time adapted to a given intensity of salinity intrusion (…)"

- P1,L23: please specify "unprecedented high salinity intrusion"; i.e. was the 2015/2016 event characterised by the time duration of salinity issues, or rather its concentration, or the groundwater depth in which salt water was found, or measured in terms of agricultural losses, etc. etc.

- P1,L29: the current figure fails to show what are "coastal areas of the delta" or rather land-locked areas (also see multiple comments posted in the PDF version of Figure 1). Please modify the map accordingly.

- P1,L29: please provide the percentage of this economic damage in respect to total value of national agricultural production for reference.

- P2,L35: is terming saltwater intrusion as agricultural drought an original idea by the authors, or has this been defined as such before by the scientific community (if the latter, please provide appropriate referencing).

- P2.L40: please clarify what type of "flow" data is required for these hydrological models.

- P2.L44: please clarify what is meant by "rainfall anomalies deficiencies"

- P2.L45: please rephrase the following sentence while using correct usage of verbs and grammar: "Additionally sea levels around the Mekong Delta continue to rise (Smajgl et al., 2015), thus causing increasing backwater effects restricting the discharge during the dry season and consequently promote salinity intrusion."

- P2.L54: "whereas" suggests a contradiction between the previous and following sentence parts, but this is not the case. Please rephrase.

- P2.L54: in addition, this is an unnecessarily long sentence that could easily be split into two.

- P2.L61: this statement is hydrologically disputable: if the authors are describing the long-term processes that connect monsoon rainfall and river flow that follows weeks/months later, "runoff" seems the wrong terminology. Where the latter describes the fast process of overland flow, the former is generally related to processes of infiltration, groundwater processes and surface water buffering such as retention.

- P3.L77: "The salinity intrusion in the Delta is measured by the hydro-meteorological services....": Firstly: how is this being measured (what instrumentation)? Secondly: please clarify "services" that are measuring the process.

- P3.L78: "The measurements are, however, not continuous, but typically performed for several days in a row, with some days without measurements in between." Are there any conditions that determine whether measurements are taken (such as high expected intrusion)? This may create a bias in measurements, which should be discussed by the authors.

- P3.L81: "The salinity measurements covered the time span 1996 – 2016": please specify whether this was a 100% coverage or provide another grounded estimate.

- P3.L85: what is the reference for this salinity threshold? Please clarify sources (same in L90).

- P4.L96: what do these ENSOxx indices represent, and how is the selection of these particular indices justified?

- P4.L109: please provide adequate sources for reference with regards to statistical methods applied (throughout L106-109).

- P6.L174: the meaning of the final part of the final sentence is not very evident. This vague statement contrasts the practical and specific recommendations made in the previous sentences. Please rephrase for clarity.

- Figure 1: major revision recommended with regards to style, intuitiveness and clarity, see marked manuscript document for all comments.

Technical corrections: See marked manuscript attached for any technical corrections related to wording or style.

Please also note the supplement to this comment:
https://www.nat-hazards-earth-syst-sci-discuss.net/nhess-2019-276/nhess-2019-276-RC2-supplement.pdf

**Supplement:**

[revised manuscript text omitted]

---

## Author Comment (AC2) · 20 Mar 2020

***Response to reviewer comment RC2 on*** "Brief communication: Seasonal prediction of salinity intrusion in the Mekong Delta" ***by*** Heiko Apel et al.

Reviewer comments in blue, authors response in black

**Anonymous Referee #2**

The authors provided an interesting manuscript on a topic that has strong relevance for actual societal problems in Vietnam and likely beyond. A seemingly novel method for long-term forecasting of salt water intrusion in cultivated lowland areas is presented, which could provide useful early warning information for damage control in agricultural production. Statistical tests by the authors result in good confidence of model performance, leading to recommendations for wider application. However, a precise idea of the actual added value of the proposed model is not communicated clearly enough, due to several reasons. These are discussed in detail below, but can be summarised as a lack of description of similar existing models, the description of input data used, and the limited possibility for model adoption due to a limited description of the model itself and data requirements. In addition, certain aspects of style, grammar, accuracy of statements and embedding in literature should all be improved in order to achieve an appropriate quality for scientific publishing with this high-profile journal.

Nonetheless, the reviewer believes there is strong potential in the manuscript (especially due to the apparent societal demand for the model); and as such, a major revision is recommended with strong encouragement for follow-up by the authors. In order to allow improvements on the remarks made by the reviewer, a long but practical list of suggestions are provided in this document (general and specific comments), as well as in the marked manuscript document (single-word suggestions).

We sincerely thank the reviewer for the overall general feedback of our work, and the constructive comments for improvement of the manuscript. This is highly appreciated. We reply to the reviewer comments below.

General comments: The abstract is rather concise, and although this can be appropriate regarding the total length of the article, perhaps a few pieces of information could be inserted. For instance, the authors could improve the technical aspect of the abstract by briefly describing the type of data that predictions are nested in (i.e. drought or ENSO indices), or by providing some quantification to support the claim for "high skill".

We included some quantification of the data used and quantification of skill in the abstract, while keeping its length in the limits of an NHESS brief communication (100 words). This is the new abstract:

"The Mekong Delta is the most important food production area in Vietnam, but salinity intrusion during the dry season poses a serious threat to agricultural production and livelihoods. A seasonal forecast of salinity intrusion is required in order to mitigate the negative effects. This communication presents a statistical seasonal forecast model based on Logistic Regression using either the ENSO34 index or Streamflow as predictor. The model is able to predict the salinity intrusion up to 9 months ahead with high skill (ROC scores > 0.8). The model can thus be used operationally as a basis for timely adaptation and mitigation planning."

The introduction section (chapter 1) clearly emphasises the importance of forecast models with an extended lead time (i.e. months rather than weeks). However, it is unclear whether such models already exist, and thus what is the novelty of the existing work. The authors should dedicate a few lines nested in scientific references to clarify this point, and thus to justify the relevance of their own contribution. In general, the use of literature is quite marginal in the manuscript, and embedding the proposed research in the scientific context is an integral part of scientific writing.

Thanks for pointing at the issue of novelty. In fact, publications dealing with forecasts of salinity intrusion in the Mekong Delta do not exist. This holds true for both short-term or seasonal forecasts. That was the reason for the missing references. The presented study is thus a novel work in this regional context, but also beyond. Publications about forecast models of salinity intrusion are rather scarce in general. There are just a few papers dealing with this issue. All of them apply different methods to the method presented in this study. The approaches of other studies are a) hydrodynamic modelling of salinity intrusion and the coupling of these models with meteorological and tidal forecast models (Risley et al., 1993), b) the use of Artificial Neural Networks (Lu and Chen, 2010;Roehl Jr. et al., 2013), c) kernel-based support vector machine (Rohmer and Brisset, 2017), and d) power law models derived from hydrodynamic models (Etemad-Shahidi et al., 2008). Logistic regression has to our knowledge never been used for salinity intrusion forecasts. Moreover, a seasonal forecast of salinity intrusion, i.e. a forecast with several months lead time, has not been published before. The presented work is thus also novel in this aspect. The use of ENSO as direct predictor also seems to be a novel aspect of the work, because relevant publication were not found. In the revised version of the manuscript, we will clearly underline these novelty aspects of the work. However, we will also take care to keep the number of citations in the allowed maximum number for a brief communication.

But besides publications in scientific literature, salinity forecasts are performed in Vietnam on an operational basis by the National Centre for Hydro-Meteorological Forecasting (NCHMF), and the Southern Institute of Water Resources Research (SIWRR). Both forecasts are based on a chain of hydrological and hydraulic models, which are fed by precipitation and tidal forecasts. The forecasts of NCHMF are short termed, i.e. with a lead time of 10 days. The forecasts of SIWRR cover also longer lead times up to a maximum of two months. The core of the forecast model of SIWRR is described in Toan (2014), but not the operational application. Because of the longer lead times and the low data requirements, the presented method is a valuable addition to the operational salinity forecasts, and can be easily ingested into the operational forecast schemes. Note that two of the co-authors are the responsible persons for the salinity forecasts at NCHMF and SIWRR, therefor a clear statement can be made in this regard.

In the meantime, the proposed forecast model has been tested by forecasting the salinity intrusion in the dry season of 2019-2020.  Using the ENSO34 index of April 2019 the model predicted a probability of exceedance of 0.98 for the 3 g/l threshold of the mean salinity in February-March 2020, and a probability of 0.8 for an exceedance of the 4 g/l threshold. The forecast after the flood season using SSI3 in November and December gave probability of exceedance of the 4 g/l threshold of 0.95. This means the model predicted a severe salinity intrusion with high confidence. The actual salinity intrusion in the Mekong Delta is currently (start of March 2020) indeed very high, reaching levels as high or even more extreme than during the record salinity intrusion of 2015-2016 (MARD, 2020;UN Vietnam, 2020). This means that the current salinity intrusion in the Mekong delta could have been forecasted about nine months in advance. We suggest to add a paragraph highlighting this model validation in the discussion section.

The description of methodology (chapter 2) deserves some critical attention to ensure an appropriate description of processes, used data, and analysis methods. The reviewer

refers to the marked manuscript as well as the specific comments provided below for all of these points.

This point was also raised by reviewer 1. We will elaborate the method and the required data in more detail in the revised manuscript. The description of the method in the main text of the manuscript will be extended as follows, accompanied with some analytical plots/data analysis in the foreseen supplement.

Method description:

[revised manuscript text omitted]

With regards to the results (chapter 3), the authors seem to present a robust set of statistical testing for findings optimal predictors. In the final lines of this chapter, an interesting point is made about the validity of ENSO-based predictions (optimal on long-term) and SSI-based predictions (optimal for short-term). Was any performance testing done where the two indices were combined, as an "optimised predictor"? If not, the authors may discuss the possibilities for this in future explorations. In addition, chapter 3 in its current form does not provide any discussion with regards to previous scientific works (e.g. regarding other long-term forecasting models), but is mostly restricted to "results".

Preliminary tests have been conducted using both indices at the same time in a Multinomial Logistic Regression as a forecast model. The results did not improve compared to the presented models. This has two reasons:

1. The forecast were already quite good with the single predictors for long-term and short-term forecasts. Therefor there was little room for improvement.

2. The less performing forecasts during the flood seasons did not improve substantially. This can be explained by the hydrological rational outlined in the manuscript. The ENSO index has a meaning for the monsoon strength and thus the discharge during the flood and following dry season only in the months before the monsoon/flood season. During the monsoon season the actual state of the ENSO looses importance for the ongoing monsoon. The discharge measured after the flood season is on the contrary a good indicator for the overall dry season flow and thus salinity intrusion. During the flood season both predictors have limited predictive power for the dry season discharge, therefore a combination does not improve the forecast.

Following this rational, a third predictor indicating the rainfall over the Mekong basin would likely have better predictive power then ESNO and SSI during the flood season. Therefor we argue that a reasonable mid-term forecast of salinity intrusion in the dry season is likely best achieved using rainfall sums over the Mekong basin over the monsoon season, i.e. June to September/October. However, in order to provide forecast in a timely manner, a near-real time rainfall monitoring product should be used for this. The telemetric ground-based rainfall monitoring network of the Mekong River Commission could be one option, or a near-real time satellite product. The TRMM-based Multi-satellite Precipitation Analysis (TMPA / 3B4x) with its latency of 1-2 month might be an option worth testing. We will include this point of extension of the forecast model in the discussion.

In the conclusions section, potential application of the proposed model is well described and its wider use is encouraged. However, the requirements with regards to data availability are not entire clear. The authors mention that data availability should be "sufficient", but do not specify or quantify what is the required coverage of flow data and the expected impact on prediction accuracy. This actually links back to the method- ology section of the

manuscript, where a quantification of data coverage in the presented study is missing as well. More clarity is required on this topic, both in described methodology and in recommendations for future applications.

We will be more specific with regards to the data requirements and transferability of the model. In short, in order to transfer the model to different locations, a continuous time series of salinity and discharge measurement is required (the ENSO index is readily available from public sources). The length of the time series should be sufficiently long for robust model fitting. There is no general rule for a sufficient length, but based on personal experience a time series covering a minimum of 15 dry season should be used. If it is less, the chances for model overfitting and spurious results are quite high. This recommendation will be added to the conclusion section.

Specific comments:

- Page1,Line7: While acknowledging the Mekong Delta as the most important Vietnamese food production area, the value of this zone with regards to agriculture and food security could be more strongly emphasised by adding two pieces of information: (1) the fraction of rice production out of total (staple) food production in Vietnam; (2) the importance of "nationally produced" food vs. imported food with regards to food security (or possibly exported value). A second line including such information would create a more solid argument as to the context of salt water intrusion and its negative impacts.

The Mekong delta is, as mentioned, the key food production area of Vietnam. In 2018, rice production was about 23.5 million tons (56% of the total production in Vietnam), 0.673 million tons of shrimp (70% of Vietnam total), 1.41 million tons of catfish (95% of Vietnam total) and 4.3 million tons of fruits (60% of Vietnam total). The total agriculture export value of the Mekong delta amounts to 8.43 Billion US dollars. This is 20% of the overall total agricultural exports of Vietnam (Son, 2020). We will add these information to the revised manuscript.

- P1,L19: Is indeed the frequency of droughts, rather than the likely duration of the most severe drought (period), the major manifestation of climate-induced intrusion?

It is both. The severeness as expressed by longer durations and higher intensities (i.e. low discharges and higher salinity levels in this context), as well as the higher frequency of droughts/salinity intrusion. Both factors are a consequence of sea level rise and climate change. In order to emphasis this, the sentence is changed to:

"Sea level rise and climate change aggravate this problem causing more severe, longer lasting, and more frequent droughts, with the consequence of more severe (longer lasting and higher salinity levels) and more frequent salinity intrusions during the dry season…"

- P1,L20: "... agricultural production system and peoples livelihood developed over historical periods and thus adapted to normal intensity of salinity intrusion (. . .)". This sentence reads slightly unclear and could likely be simplified, e.g. as ". . .agricultural production systems and livelihoods over time adapted to a given intensity of salinity intrusion (. . .)"

Changed as suggested.

- P1,L23: please specify "unprecedented high salinity intrusion"; i.e. was the 2015/2016 event characterised by the time duration of salinity issues, or rather its concentration, or the groundwater depth in which salt water was found, or measured in terms of agricultural losses, etc. etc.

"Unprecedented high salinity intrusion" is defined by salinity levels, duration and extent of salinity intrusion. In order to clarify this, the sentence will be changed to:

"This unprecedented severe salinity intrusion, which manifested by the earliest onset of high salinity levels, the highest observed salinity measurements in most estuaries of the Mekong, and the longest duration and the deepest penetration of saline water in the river system ever observed, caused...."

- P1,L29: the current figure fails to show what are "coastal areas of the delta" or rather land-locked areas (also see multiple comments posted in the PDF version of Figure 1). Please modify the map accordingly.

Figure 1 was completely re-worked. See the new figure in the answer to the specific comments on Figure 1 below.

- P1,L29: please provide the percentage of this economic damage in respect to total value of national agricultural production for reference.

We set the damage in relation to the national GDP of the agriculture, fishery and forestry sectors, as provided by the General Statistics Office of Vietnam. The damage accounts for 0.74% of the GDP in these sectors in 2016.

- P2,L35: is terming saltwater intrusion as agricultural drought an original idea by the authors, or has this been defined as such before by the scientific community (if the latter, please provide appropriate referencing).

To our knowledge this is an original definition by us. There are a few standardized drought indices available, but all refer to physical water availability, not water quality. Other studies used additional factors for defining agricultural droughts, but we could not find a study using salinity of irrigation water for a drought definition. In order to make this clear we rephrased the text as follows:

"Therefor salinity intrusion in the Mekong Delta can be termed as agricultural drought. The general definition of agricultural drought is a situation, in which plant water demands cannot be satisfied by water availability (Mannocchi et al., 2004;Mishra and Singh, 2010). By terming salinity intrusion an agricultural drought we hereby extend the definition of water availability from a pure physical, quantitative view to a water quality perspective. This agricultural drought is a serious hazard for large parts of the population, for which agriculture is still the basis of its livelihood."

- P2.L40: please clarify what type of "flow" data is required for these hydrological models.

Flow means daily river discharges. The text is changed accordingly.

- P2.L44: please clarify what is meant by "rainfall anomalies deficiencies"

This is an error, thanks for spotting it. Negative rainfall anomalies (or rainfall deficiencies) is meant. The sentence is changed to:

"A drought such as in 2016 is expected to occur more often in future, as negative rainfall anomalies occurring with El Nino events are expected to occur more frequently..."

- P2.L45: please rephrase the following sentence while using correct usage of verbs and grammar: "Additionally sea levels around the Mekong Delta continue to rise (Smajgl et al., 2015), thus causing increasing backwater effects restricting the discharge during the dry

season and consequently promote salinity intrusion."

The sentence is changed to:

"Moreover, sea levels around the Mekong Delta are rising and are expected to rise further in future (Smajgl et al., 2015). Rising sea levels cause increasing backwater effects in the river channels, and thus promote salinity intrusion during the dry season."

- P2.L54: "whereas" suggests a contradiction between the previous and following sentence parts, but this is not the case. Please rephrase.

The sentence is changed to:

"The climate and hydrology of the Mekong Delta and Mekong basin are dominated by the monsoonal climate, separating the hydrological year into distinct rainy/high flow and dry/low flow seasons. The hydrological regime lags the climate regime depending on the location in the basin."

- P2.L54: in addition, this is an unnecessarily long sentence that could easily be split into two.

See answer above.

- P2.L61: this statement is hydrologically disputable: if the authors are describing the long-term processes that connect monsoon rainfall and river flow that follows weeks/months later, "runoff" seems the wrong terminology. Where the latter describes the fast process of overland flow, the former is generally related to processes of infiltration, groundwater processes and surface water buffering such as retention.

In order to avoid definition problems, the sentence is re-phrased to:

"In the delta of the Mekong this lag is most noticeable due to the time required for transforming rainfall in the about 800,000 km$^2$ large basin to river discharge and routing the discharge to the delta."

- P3.L77: "The salinity intrusion in the Delta is measured by the hydro-meteorological services. . .": Firstly: how is this being measured (what instrumentation)? Secondly: please clarify "services" that are measuring the process.

The salinity is measured by collecting water samples, which are then analyzed in the laboratory. The "services" we refer to is actually the Southern Regional Hydro-Meteorological Center (SRHMC), which is the official agency for collecting this data. We will change the sentence accordingly in order to avoid misunderstanding.

- P3.L78: "The measurements are, however, not continuous, but typically performed for several days in a row, with some days without measurements in between." Are there any conditions that determine whether measurements are taken (such as high expected intrusion)? This may create a bias in measurements, which should be discussed by the authors.

Salinity measurements are performed at 39 locations in the Mekong Delta during the dry season, starting in January until June (until 2013 the measurements started in February). The measurements are taken in mid-river at 0.2, 0.5 and 0.8 of the water depth. The reported salinity is the mean of these three measurements. If the water depth is below 3 m, only one sample is taken at 0.5 of the depth. Due to constraints in personal and monetary resources,

the monitoring is, however, not time continuous. The general scheme is to measure 2-3 days in a row at 2 hours intervals (i.e. 12 measurements per day). This measurement period is followed by a 2-4 days without measurements, after which the monitoring resumes. The exact monitoring scheme is defined for each monitoring location individually, depending on the hydrodynamics and the tidal regime. Because of the different sampling schemes per station (and possibly also per year) it is difficult to describe the monitoring and the resulting salinity time series in a general way. In the revised manuscript we would thus present the salinity time series of the gauging station Son Doc, which was used for the development of the forecast models.

The following figure shows the time coverage of salinity measurements at Son Doc:

[Figure]

Every dot represents a day with measurements, i.e. 12 samples at 2-hour intervals. The figure shows that the measurements are almost similarly distributed during February and March (the gray shaded area) in the different years. The mean salinity in February and March, which is the predictand of the forecast model, is calculated from these 2-hourly measurements.

The mean number of samples in February and March from 1997 - 2016 is 310 samples with a standard deviation of 38. These sample statistics along with the similarity of the temporal sampling shown in the figure above means that the sampling schemes (in terms of numbers and schedule) of the individual years are well comparable, and a bias caused by the sampling scheme is unlikely. The exception is the year 1996, which has continuous measurements without breaks. In order to test a possible bias in the 1996 data in relation to the other years, the 1996 data was resampled 1000 times with the mean number of samples of the other years (i.e. 312 samples of 719 samples in February-March 1996). The resampled mean FebMar salinity in 1996 is 3.28 g/l, which is practically identical to the mean of all samples (3.27 g/l). Therefor we can reject the assumption of a bias in the data introduced by the sampling scheme.

We will explain the sampling in short in the main part of the revised manuscript, and add the figure and statistics outlined above in the supplement.

- P3.L81: "The salinity measurements covered the time span 1996 – 2016": please specify whether this was a 100% coverage or provide another grounded estimate.

Yes, the time series is continuous, i.e. it covers all dry seasons in the time span 1996 – 2016. An according statement was inserted in the paragraph. Details of the sampling are provided in the reply to the previous comment, and will be added in a supplement to the manuscript.

- P3.L85: what is the reference for this salinity threshold? Please clarify sources (same in L90).

This threshold is used by the authorities in Vietnam for warnings of a severe salinity intrusion. It is always stated in governmental reports of salinity intrusion. Therefor this threshold was primarily selected, in order to enable a comparison with official short term forecast and to raise the acceptability of the model within the governmental agencies of Vietnam.

Moreover, the 4 g/l threshold can be correlated to studies showing significant decreases in crop yield or even crop failures with higher salinity levels (Grattan et al., 2002;Kotera et al., 2014;Zeng and Shannon, 2000;Zeng et al., 2001). In order to clarify this, the sentence is changed as follows, including a reference to a study analyzing the effect of salinity on rice crop growth and yield:

"For paddy rice a salinity of the irrigation water exceeding 4 g/l is generally regarded as too high for the plants to survive during the vegetative stage by the authorities in Vietnam (compare Zeng and Shannon, 2000)."

- P4.L96: what do these ENSOxx indices represent, and how is the selection of these particular indices justified?

We added some lines explaining the different ENSO indexes. However, we will not provide an extended explanation and discussion of the indexes, because they are described in numerous papers and even in many internet resources, e.g. from the US National Oceanic and Atmospheric Administration (NOAA). The text section will be changed to:

"The ENSO indexes tested were monthly ENSO1, ENSO3, ENSO4, and ENSO34 indexes. All of these indexes aim at representing the state of the El Niño Southern Oscillation by considering sea surface temperatures at different regions of the Pacific Ocean, whereas ENSO34 is regarded as the most appropriate sea surface temperature index representing the general state of the ENSO (Bamston et al., 1997). The testing of different indexes aims at the identification of the most robust predictor for salinity intrusion in the MKD. All of the ENSO indexes used in the forecast model start in April of the year before the considered dry season, i.e. with a lead time of up to 9 months before the start of the forecasted FebMar time period."

- P4.L109: please provide adequate sources for reference with regards to statistical methods applied (throughout L106-109).

The methods and goodness-of-fit measures are very standard statistical methods and can be found in any good statistical handbook. However, we added references as examples, as already listed in the reply to the request for an extension of the method description above. The references to the performance criteria read now:

"One model was fitted for all predictors and lead times, and the best performing ENSO and SSI predictors for the different lead times were manually selected according to the following criteria:

- The Receiver Operator Characteristic (ROC) score (Mason, 2008).

- The Akaike Information Criteria (AIC) (Burnham and Anderson, 2004).

- The Cragg and Uhlers (also known as Nagelkerke) Pseudo-$R^2$ (Nagelkerke, 1991), which is defined for categorical variables analogously to the normal $R^2$ for continuous variables.

- The accuracy (rate of correct forecasts)."

- P6.L174: the meaning of the final part of the final sentence is not very evident. This vague statement contrasts the practical and specific recommendations made in the previous sentences. Please rephrase for clarity.

We will rephrase the final sentence to: "Mitigation actions should include negotiations with the riparian countries aiming at an adaptation of the operation schedule of reservoirs in the Mekong basin in order to maintain sufficient flow during the dry season."

We will, however, change the context of the sentence, because the statements before about the expected severe salinity intrusion are already outdated, as the dry season 2020 is already ongoing. We suggest, as stated in a previous reply above, to include the model forecast of the dry season 2020 and the ongoing salinity intrusion as model validation. The statement of early negotiations among the Mekong riparian countries based on the long-term forecast will follow this validation.

- Figure 1: major revision recommended with regards to style, intuitiveness and clarity, see marked manuscript document for all comments.

Many thanks for the thorough and critical review of the figure. The comments were all ingested, resulting in the new Figure 1 shown below:

[Figure]

**Figure 1: Overview map of the study area Mekong Delta. Top left: Regional overview showing South-East Asia with the Mekong basin and delta, and the neighbouring countries. The background country, ocean and topography maps are made with Natural Earth (Free vector and raster map data @ naturalearthdata.com). Bottom: The Vietnamese part of the Mekong Delta with the location of the main official and permanent hydro-meteorological monitoring stations and the salinity monitoring station Son Doc. The land use map of the Mekong Delta as in 2010 is shown as reference illustrating the different land use types in the different regions of the delta. The map was derived at 500m resolution from Moderate Resolution Imaging Spectrometer satellite (MODIS) images (Source: Catch-Mekong Knowledge Hub, https://catchmekong.eoc.dlr.de/Elvis/, provided by the German Aerospace Center DLR. The method of land use classification is described in Leinenkugel et al. (2013)).**

Technical corrections: See marked manuscript attached for any technical corrections related to wording or style.

All the comments will be considered in the revised manuscript.

---

## Author Response (AR2)

Reply to reviewer comments and editor comments to the revised manuscript

**Brief communication: Seasonal prediction of salinity intrusion in the Mekong Delta**

By

Heiko Apel, Mai Khiem, Nguyen Hong Quan, To Quang Toan

We would like to thanks reviewer #2 for his positive feedback on our revised manuscript. We revised the wording of the last section in the conclusions and checked the grammar of the whole MS again. At this point we would like to mention our high appreciation of the first constructive review provides by the reviewer, which helped to improve the MS a lot.

We would also like to thank the editor Paolo Tarolli for his positive feedback. We changed the figures according to his advice.

With kind regards on behalf of all authors,

Heiko Apel

Potsdam, April 30th 220